# The Role of Krüppel-like Factors in Pancreatic Physiology and Pathophysiology

**DOI:** 10.3390/ijms24108589

**Published:** 2023-05-11

**Authors:** Michael Giarrizzo, Joseph F. LaComb, Agnieszka B. Bialkowska

**Affiliations:** Department of Medicine, Renaissance School of Medicine at Stony Brook University, Stony Brook, NY 11794, USA; michael.giarrizzo@stonybrookmedicine.edu (M.G.); joseph.lacomb@stonybrookmedicine.edu (J.F.L.)

**Keywords:** Krüppel-like factors, pancreatitis, pancreatic cancer, metabolism, stemness, reprograming

## Abstract

Krüppel-like factors (KLFs) belong to the family of transcription factors with three highly conserved zinc finger domains in the C-terminus. They regulate homeostasis, development, and disease progression in many tissues. It has been shown that KLFs play an essential role in the endocrine and exocrine compartments of the pancreas. They are necessary to maintain glucose homeostasis and have been implicated in the development of diabetes. Furthermore, they can be a vital tool in enabling pancreas regeneration and disease modeling. Finally, the KLF family contains proteins that act as tumor suppressors and oncogenes. A subset of members has a biphasic function, being upregulated in the early stages of oncogenesis and stimulating its progression and downregulated in the late stages to allow for tumor dissemination. Here, we describe KLFs’ function in pancreatic physiology and pathophysiology.

## 1. Introduction

The pancreatic parenchyma is divided into exocrine and endocrine sections, both originating from the outgrowth of the primitive gut. The exocrine portion is comprised of acinar and ductal cells and amounts to 96–99% of the total pancreas, and the endocrine islet of Langerhans comprises 1–4% of the entire pancreas [1,2]. Digestive enzymes are synthesized in acinar cells and transported to the gastrointestinal tract via ducts. Neuronal and hormonal signals regulate the secretion rate. At the same time, the endocrine gland consists of five secretory islet cells and is mainly responsible for maintaining glucose homeostasis [3,4]. Thus, dysregulation of pathways that regulate functions of both compartments due to injury could impact their homeostasis and result in disease development and progression. Krüppel-like factors (KLFs) belong to the family of transcription factors characterized by highly conserved three zinc finger domains in their C-terminal portions [5,6,7]. To date, seventeen KLFs have been identified and studied in the context of various diseases. KLFs are expressed in multiple organs and tissues and have been studied in homeostasis, disease development, and progression. KLFs have been shown to regulate cell proliferation, apoptosis, migration, invasion, stem cell self-renewal, differentiation, and survival. Comprehensive reviews pertaining to their structure and function have been published elsewhere [7,8,9,10,11,12,13].

Interestingly, KLFs can exert opposite effects in regulating the same pathways and initiate different cell responses, as some of them can function as tumor suppressors (KLF2, KLF8) and some as oncogenes (KLF7, KLF9). In addition, the same KLF can be a suppressor or an oncogene in a different type of cancer. Curiously, depending on the disease stage, as in the case of KLF4 and KLF5, their role can switch between oncogene and tumor suppressor. Recent years brought a better understanding of their role in pancreatic pathophysiology, specifically in the context of islet function, acute injury to the pancreas, and their involvement in pancreatic tumorigenesis. Here, we present a summary of the current understanding of KLFs’ role in the endocrine compartment of the pancreas, regenerative capabilities, and their impact on pancreatic cancer development and progression.

## 2. Krüppel-like Factors in Regulation of Islet Function

### 2.1. Homeostatic Pancreatic Endocrine Function

Pancreatic endocrine function occurs in regions known as pancreatic islets or islets of Langerhans. These islets, while only 1–4% of the total pancreatic volume, are critical regulators of glucose metabolism. Pancreatic islet β-cells produce and secrete insulin and are typically 50–75% of cells in a pancreatic islet [14]. While the mechanism is not well understood, transforming growth factor-β (TGFβ) signaling regulates pancreatic islet formation and mass. KLF10 and KLF11 are transcription factors controlled by TGFβ. KLF10 activates SERTA domain-containing 1 (SEI-1) transcription, in turn, upregulating p21^Cip1^ expression which induces cell cycle arrest by inhibiting cyclin-dependent kinase 4 (CDK4)/cyclin-D complexes [15,16]. In vivo knockout of *KLF10* reduces murine islet mass and increases glucose intolerance [15]. KLF11 directly activates the insulin promoter in β-cells in high-glucose conditions [17]. Other studies, however, demonstrated suppressed KLF11 expression in β-cells in high glucose levels. KLF11 is unable to bind to the mutant CACCC box element of the *Insulin-2* (*Ins2*) gene promoter [18]. Indeed, follow-up studies determine that KLF11 selectively regulates insulin promoter (InsP) transcription of β-cells depending on interactions with the A3 and CACCC box elements and the presence of p300 and Pancreatic and duodenal homeobox 1 (PDX1) cofactors [19,20,21]. Additional studies are needed to understand the effect, if any, that KLF10 and KLF11 have on β-cell volume, function, and metabolic homeostasis.

Pancreatic endocrine function is also moderated by systemic feedback. Mice with *Klf15* overexpression in white and brown adipose tissue exhibited increased pancreatic β-cell insulin secretion, greater insulin resistance, and decreased diet-induced obesity. Stearoyl-CoA desaturase 1 (SCD1) expression and total oxidative stress levels were also inhibited [22]. Altogether, it is implied that KLF15 in adipose tissue regulates insulin secretion and resistance via reduced SCD1 and oxidative stress.

### 2.2. Type 1 Diabetes

Globally, diabetes mellitus (DM) is one of the most common metabolic diseases. The three main types are type 1 diabetes, type 2 diabetes, and gestational diabetes (during pregnancy). Type 1 diabetes (T1D) is characterized as the inability to produce and secrete insulin predominantly due to autoimmune T-cell-mediated destruction of islet β-cells. During disease onset, the autoimmune T-cell population is enriched by C-X-C chemokine receptor type 5 (CXCR5)^+^CD4^+^ T follicular helper (TFH) cells [23]. This TFH phenotype is mediated by miR-92a and induced via phosphatase and tension homolog (PTEN)–phosphoinositol-3-kinase (PI3K) signaling [23]. KLF2, a regulator of lymphocyte migration homing receptors, TFH generation, and B-cell priming, is downregulated in CD4^+^ T-cells during islet autoimmunity [23,24,25,26]. miR-92a directly inhibits *KLF2* whereas a miR-92a antagomir reduces TFH precursor induction [23,27] (Figure 1A). Therapeutic targeting of miR-92a or the PTEN–PI3K–KLF2 pathway may rescue islets from aberrant immune response during T1D autoimmunity.

Muscle wasting, the loss of skeletal mass and function, is common in patients with diabetes [28,29] and is caused by aberrant protein synthesis and degradation [30]. Protein degradation is controlled by several proteolytic pathways including the ubiquitin–proteasome system (UPS), lysosomal autophagy, capase-3, and calpain systems [31,32,33]. Ethanolic extract of *Schisandrae chinensis Fructus* (SFe), used in traditional medicine to treat a range of diseases and disorders, decreased atrophic factors while increasing muscle mass and strength in streptozotocin-induced diabetic mice without altering blood glucose levels. Molecularly, the UPS-regulating proteins KLF15 and p-CAMP responsive element-binding protein (p-CREB) were reduced in SFe-treated diabetic mice [34]. SFe may have therapeutic potential to prevent muscle wasting in diabetic patients by restoring homeostatic protein synthesis-degradation balance via the CREB–KLF15–UPS pathway.

A paradigm of two macrophage subtypes, M1 and M2, regulate inflammatory response. M1 macrophages are pro-inflammatory and usually the first immune cells to infiltrate pancreatic islets. Conversely, M2 macrophages are anti-inflammatory and promote survival, wound healing, and regeneration [35]. An imbalance favoring M1 macrophage response contributes to autoimmunity in T1D. KLF4 regulates differentiation and epithelial-to-mesenchymal (EMT) transition in M2 macrophages [36] (Figure 1B). Furthermore, reduced KLF4 suppressed myeloid-derived suppressor and fibrocyte populations and impaired wound healing [37,38]. Accumulation of infiltrative KLF4^+^ M2-like mesenchymal cells and fibroblasts exhibit protective effects of pancreatic islets from T1D autoimmunity [39,40]. The mechanism by which KLF4 is inhibited is currently unknown. Targeting of KLF4 in these macrophage populations may restore homeostatic M1/M2 populations and, subsequently, preserve pancreatic islets during T1D onset.

Developing models, such as inducible pluripotent stem cells (iPSCs), is critical to studying disease. Recently, adult skin fibroblasts from T1D patients were transformed using mRNAs encoding octamer-binding protein 4 (OCT4), SRY-box transcription factor 2 (SOX2), KLF4, MYC proto-oncogene, BHLH transcription factor (c-MYC), and lin-28 homolog A (LIN28) transcription factors. The resultant iPSCs exhibited human embryonic stem-cell-like features and functional expression of pancreas-specific miRNA [41]. This methodology can serve to generate both models of disease and cell-based therapeutic technology. In Section 3, we describe the latest techniques employed to regenerate the pancreas.

### 2.3. Type 2 Diabetes

Type 2 diabetes (T2D), the prevailing diabetes type, is characterized by insulin resistance and β-cell dysfunction. As with many metabolic diseases, T2D onset is associated with aging, among other factors. A recent meta-analysis combined single-cell transcriptomics, transcription factor regulation, microscopy, and insulin secretion datasets to model the effect of aging on β-cells. Transcription factor activity decreased while endoplasmic reticulum stress increased over time, resulting in upregulated β-cell autophagy and molecular heterogeneity. Ultimately, β-cell aging compromises number and function leading to T2D onset [42].

KLF6 promotes β-cell proliferation and regulates β-cell de- and transdifferentiation into glucagon-producing α-cells, whereas in vivo knockdown of *Klf6* has the opposite effect on β-cell proliferation and de/transdifferentiation in the setting of gestational, diet-induced, and insulin receptor antagonist-induced insulin resistance [43] (Figure 1C). KLF7 is expressed in many human tissues including pancreas, liver, skeletal muscle, and adipose, and cell lines including adipocytes and β-cells [44,45]. In vitro overexpression of *KLF7* in β-cells inhibited expression of the glucose transporter solute carrier family 2 member 2 (SLC2A2) and the ATP-sensitive potassium channel subunits ATP-binding cassette subfamily C member 8 (SUR1) and potassium inwardly rectifying channel subfamily J member 11 (Kir6.2), known to regulate cellular metabolism and insulin secretion [45] (Figure 1D). This inhibitory mechanism of action must be further investigated to better understand the role *KLF7* has in T2D pathogenesis.

Several studies investigated T2D risk variants at known susceptibility loci. Analysis grouped risk loci into five major categories: insulin sensitivity/resistance, insulin secretion and fasting hyperglycemia, insulin processing defects, insulin secretion without change in fasting glucose levels, and no defined associations to glycemic traits. KLF11 variants inhibit insulin transcription, resulting in lower β-cell insulin production and secretion [17] (Figure 1E). KLF14 is a locus of the insulin sensitivity/resistance cluster and polymorphic variants are associated with T2D [46,47] (Figure 1F). An increase in age-related KLF14 DNA methylation has been observed in pancreatic islets. Moreover, epigenetic methylation in blood is associated with such changes in islets, suggesting that circulating biomarkers may be useful in diagnosing and tracking disease progression [48].

## 3. Regenerative Capabilities of Krüppel-like Factors

Discovering novel therapeutics to overcome the lack of treatment options and limitations of organ transplants for pancreatic cancer and other pancreatic diseases such as diabetes is crucial to saving lives. Therefore, researchers pursued alternatives to small molecule therapies by bioengineering pluripotent, self-regenerative cells that restore normal function to damaged tissue. By manipulating the natural genetic induction of pluripotent activity, iPSCs are generated through reprogramming of adult fibroblasts and tissue-specific cells using a defined set of transcriptional and growth factors, including KLF4 [49]. iPSCs have the potential to emulate natural physiological conditions and offer autologous regenerative medicines with limitless transdifferentiation, while circumventing critical safety concerns such as insertional mutagenesis and graft vs. host disease [50]. However, current obstacles remain as establishing robust efficiency for generating pure populations of specific differentiated cell types has proven difficult and uncontrolled iPSC growth can lead to disorganized teratoma formation [51,52].

KLF4, a chromatin reorganizer and enhancer of transcriptional rewiring, has been utilized as an influential proliferative and differentiation factor in reprogramming terminally differentiated somatic cells to generate pluripotent stem cells in vitro with great success, albeit low efficiency [50]. The diverse ability of KLF4 to activate genes involved in regulating cell-cycle and its essential involvement in embryonic development and adult stem cell activity has made it an important candidate for functional reprogramming [53]. KLF4, OCT4, SOX2, and c-MYC, collectively, comprise a group of DNA-controlling proteins called the Yamanaka factors, which are highly expressed in embryonic stem cells (ESCs) during primordial germ layer development. These factors shut off transcriptional processes for cell-specific gene expression which suppresses the characteristic properties that define a cell type, and then activate transcription of pluripotency genes resulting in dedifferentiation and development of stem cell characteristics [54]. By defying the concept of irreversible commitment once a cell differentiates, the Yamanaka factors give researchers a replicable method for developing treatments based on redefining cellular lineage.

Type 1 diabetes (T1D) and pancreatic ductal adenocarcinoma (PDAC) are two pancreatic diseases that affect millions worldwide annually. PDAC is projected to be the second leading cause of cancer mortalities by 2030, because effective screening and early diagnosis are unavailable [55]. The current source of cell replacement for T1D patients with damaged β-cells is donor organs that are limited, irregular, and require lifelong immunosuppressive co-treatment to prevent transplant rejection [56]. Generating physiologically representative pancreatic cancer models using iPSCs may develop novel diagnostic and treatment strategies [57]. Moreover, regenerating β-cells from autologous in vitro reprogramming is an attractive method for restoring damaged islet cell mass in T1D patients [50].

### 3.1. iPSCs from Non-Pancreatic-Specific Cell Types

Functioning as a regenerative unit of the skin wound healing and recovery process, human dermal fibroblasts are a popular somatic cell type for induced reprogramming. They are an easily accessible cell source with a high proliferative capacity, and large amounts of autologous cells can be collected and cultured using minimally invasive methods [58]. Through a rudimentary protocol, iPSCs generated from fibroblast cells were differentiated into insulin-producing islet-like clusters that contained cells positive for glucagon and c-peptide, two characteristic biomarkers of pancreatic α- and β-cells, respectively (Table 1) [49]. Upon extracellular glucose stimulation, the positive cells released c-peptide, which provided early evidence for the generation of functional populations of pancreatic cells derived from somatic cells as a treatment for diabetes [49]. However, the general efficiency for iPSC differentiation was very low, and c-peptide levels secreted by iPSC β-cells were much lower than those produced by adult human β-cells [49].

Researchers later identified higher reprogramming efficiency in keratinocytes, the most dominant skin cell type, similar to dermal fibroblasts in their essential regenerative functions. Keratinocytes are more amendable to reprogramming due to their transcriptional profile being far similar to human ESCs than fibroblasts [59]. Keratinocyte-derived iPSCs (KiPSC) underwent similar genetic changes to ESCs in response to in vitro developmental cues and were observed to be indistinguishable in morphology, growth properties, and expression of pluripotent biomarkers [59]. These extrinsic factors upregulated forkhead box A2 (FOXA2) and hepatocyte nuclear factor 4 alpha (HNF-4α) expression, primitive gut tube markers, as well as Haematopoietically expressed homeobox (HEX) and ISL LIM homeobox 1 (ISL1), which are indicative of pancreatic endoderm expression (Figure 2). Overall, keratinocytes were observed to yield iPSC colonies in half the time (10 days) that fibroblasts were shown to (20+ days), and their reprogramming efficiency approached 1% of the total cells, 100-fold higher than that reported for fibroblasts [59].

As more defined induction protocols were developed, in vivo experiments further built upon our understanding of iPSC characteristics. Transgenic mice were treated to develop in vivo teratomas emerging from multiple organs (Table 1). Yamanaka-induced iPSCs continued to proliferate independently after treatment conditions were removed [53]. When extracted, nearly 71% of the developed teratomas were well-differentiated with abundant cell-type derivatives from all three embryonic germ layers [52]. Moreover, in-vitro-generated iPSCs injected intraperitoneally into mice yielded embryo-like structures expressing lineage-specific markers, including SOX2 for ectoderm, Brachyury (T) for mesoderm, and transcription factor GATA-4 (GATA4) for endoderm [52]. Teratoma formation introduced foundational safety concerns regarding disorganized tumor potential. However, iPSCs displayed remarkable diversity and plasticity in vivo by differentiating into various cell types and generating embryonic pluripotency characteristics typically absent in an adult organism [52].

To narrow our understanding of the independent factors that drive pluripotency, another study knocked out insulin receptors (IR) in mice to disrupt IR-mediated signaling in pluripotency and lineage development. The loss of insulin receptors produced a compensatory effect of upregulated pluripotency marker expression, including KLF4, which stimulated protein activity associated with self-renewal and growth in the cells [61]. Phosphoproteomic analysis of pluripotency signaling pathways found downregulated phosphosites and phosphorylation on growth regulatory proteins, including signal transducer and activator of transcription 3 (STAT3), mechanistic target of rapamycin kinase (mTOR), and mitogen-activated protein kinase 1 (ERK) [61]. Further insight into the molecular mechanisms and independent cellular factors that drive the signaling of pluripotency regulators is crucial to optimizing efficiency and creating a stable microenvironment for iPSC transplants.

### 3.2. iPSCs from Pancreatic-Lineage Specific Cell Types

Researchers began targeting tissue- and lineage-specific cells to develop cell therapies for T1D. Transplanting autologous insulin-producing cells as a functional cure would address critically unmet needs, given the current limitations of T1D treatments. Research has shown the extreme nuances and difficulties in isolating pancreatic stem cells [62]. Purification and expansion of native pancreatic stem cells have been attempted and failed because they comprise such a small proportion of the pancreatic cell population [60]. Deriving and reprogramming adult human pancreas cells expressing embryonic antigen 4, showed that tissue-specific/progenitor cells have an excellent capacity for lineage-specific differentiation, which could facilitate autologous transplantations [63]. These cells exhibited higher expression of *KLF4*, *SOX2*, *OCT4*, and homeobox transcription factor Nanog (*NANOG*) mRNA levels, suggesting greater potential as a source of inducible pancreatic lineage-specific cells compared to other previously used cell types such as fibroblasts [63]. Additionally, positive correlations have been found between the number of induced pancreatic islet-specific progenitors transplanted and the long-term post-transplant metabolic success of glucose tolerance [60].

One study generated glucose-sensitive, insulin-producing cells using mouse pancreatic cells (Table 1). These cells not only proliferated in vitro and expressed pluripotency-related gene expression but also expressed PDX1, a major biomarker of the pancreatic β-cell lineage [56]. These induced tissue-specific stem cells derived from somatic pancreas cells (iTSC-P) released insulin in a dose-dependent responsive manner to exogenous glucose stimulation in vitro, albeit with a relatively slow response compared to endogenous insulin release [56]. Additionally, around 5–6% of these differentiated cells were insulin- and c-peptide-positive, a five-fold increase compared to ESCs [50]. The crucial benefit of iTSC-P is their enhanced efficiency in differentiating and purifying, as ESCs and iPSCs are typically contaminated to differentiate and achieve higher population purity, as non-specific ESC and iPSC colonies are contaminated with disorganized cell differentiation that increases the probability of teratoma formation [50]. In vivo, iTSC-P exhibited limitless self-renewal capacity but did not develop teratomas in immunodeficient mice [56]. The physiological similarities that iTSC-P can produce compared to natural β-cells support tissue-specific cell reprogramming for transplantations. Furthermore, the ability to control and target specific tissues with pluripotent induction while eradicating any tumorigenic possibility is necessary to ensure safe treatment without causing severe and potentially lethal side effects.

Producing autologous iPSCs requires in vitro proliferation and maintenance before transplantation, however, progressive loss of intracellular insulin content is an issue that limits the potency and shelf life of these cells [64]. Microencapsulation of iPSCs within a polymeric and non-cytotoxic membrane provides an immunoprotective barrier that promotes significant growth and maturation of different cell types [64]. Furthermore, transplanting human islet-derived stem cells (hIPSCs) into non-obese diabetic (NOD) mice using a highly purified alginate-based microcapsule demonstrated niche favoring of hIPSC differentiation, which allowed for the acquisition of functionally competent β-like cell types [64].

A recent study employed a rigorously established, empirically-based protocol to stimulate in vitro differentiation of induced pancreatic-specific progenitor cells (iTPC) (Figure 2, Table 1). Of the 64 initial cell lines, 38 exhibited stem cells with similar morphology to pancreatic-specific progenitor and gut tube endodermal (GTE) cells, expressed *PDX1* mRNA, and demonstrated significant proliferative durability and longevity by lasting more than 120 days without any changes [60]. Characteristically, 16% of differentiated iTPCs were insulin and c-peptide double positive and released 2 to 3-fold higher amounts of human insulin than non-pancreatic-specific iPSC populations in response to extracellular glucose [60]. Importantly, post-transplantation analysis of differentiated iTPCs found that only 5% were positive for Marker of proliferation Ki-67 (Ki-67), indicating that proliferation amongst these cells did not present teratoma concerns and lacked initial tumorigenic markers [60]. When induced by conditions established to drive iPSCs towards other cell types such as hepatocytes and neuroectoderm, no mRNA levels characteristic of those types were subsequently detected in the iTPC population, suggesting that these cells were committed to a pancreatic lineage [60]. Overall, it became clear that tissue-specific iPSCs reprogram much more efficiently, have greater functionality, and are highly stable relative to iPSCs derived from non-specific somatic cells. Therefore, pancreatic regenerative cell therapies from induced pancreatic progenitor cells may be the most attractive method for restoring islet cell mass in T1D.

## 4. Pancreatitis and Pancreatic Cancer

Pancreatitis (acute and chronic) is a major gastrointestinal disorder, with acute pancreatitis being one of the most frequent causes of hospitalization [65,66]. Inflammation of the pancreas can be caused by numerous etiologies (i.e., alcohol, increase in enzyme production, pancreatic duct obstruction). Pancreatitis is caused by (1) acute injury to acinar cells, which is characterized by aberrant enzyme synthesis in the acinar cells, followed by inhibition of enzyme secretion, increase in inflammation, and transient damage to the tissue; (2) chronic injury, with continuous inflammation, increased activity of stellate cells, fibrosis, and atrophy of acinar cells, which can progress toward development of pancreatic intraepithelial neoplasia, a precursor to pancreatic cancer [67,68,69,70].

Injury to acinar cells due to inflammation, metabolic stress, loss of cell-to-cell interaction, or hyperactivation of Kirsten rat sarcoma viral oncogene homolog (KRAS) enables induction of acinar-to-ductal metaplasia (ADM) [71] (Figure 3). This process is characterized by a robust inflammatory response that activates intrinsic acinar signaling pathways: nuclear factor kappa B (NFkB), nuclear factor of activated T-cells 1/4 (NFATC1/4), mitogen-activated protein kinase (MAPK), STAT3, NOTCH, and transcription factor (TCF)/β-catenin [72,73,74,75,76]. It is accompanied by the re-expression of the progenitor markers that direct transdifferentiation of acinar cells to duct-like cells, including PDX1 and Hes family bHLH transcription factor 1 (HES1), as well as ductal markers such as sex-determining region Y (SRY)-box 9 (SOX9) and keratin 19 (KRT19) [77,78,79,80]. After acute injury and subsequent withdrawal of the insult, the duct-like cells revert to acinar cells, and the damage is resolved. In contrast, recurrent injury to the acinar cells due to premature activation of pancreatic enzymes causes autodigestion of the parenchyma and, subsequently, releases pro-inflammatory mediators that promote immune cell infiltration and activation of pancreatic stellate cells (PSCs). Active PSCs stimulate collagen and extracellular matrix metalloproteinase (MMP) production and synthesis of growth factors and hormones, increase susceptibility to proliferative chemokines and tissue inhibitor of metalloproteinase (TIMP), and, as a consequence, stimulate the development of fibrosis. This prolonged injury results in the irreversible dedifferentiation of acinar cells to duct-like cells that precede pancreatic intraepithelial neoplasia (mPanIN) formation, suggesting that ADM is the initial step of pancreatic cancer development [81,82,83,84].

KRAS is mutated in 90–95% of pancreatic cancers [85]. Activated KRAS expression has been shown to stimulate the generation of ADM lesions that can further progress toward mPanIN and PDAC [86,87]. However, increased expression of KRAS alone does not guarantee ADM/mPanIN formation, as only a tiny portion of acinar cells express the mutated KRAS transform. The RAS activity can be induced by extrinsic factors (e.g., inflammation), intrinsic (e.g., activating mutation in *KRAS*), or both, and only pathological/high levels of RAS activity can start ADM programming in acinar cells that increase inflammation and genetic instability required for the development of cancer. Mutations in KRAS are observed in early intraepithelial neoplasia, while *TP53*, Mothers against decapentaplegic homolog 4 (*SMAD4*), and BRCA2 DNA repair associated (*BRCA2*) happen at an advanced stage and invasive carcinomas [74,88,89,90,91,92] (Figure 3). Here, we present the role of KLFs in PDAC development and progression (Table 2).

### 4.1. KLFs as Tumor Suppressors in PDAC

#### 4.1.1. KLF2

KLF2 is a known suppressor, and its expression is lost or significantly inhibited in pancreatic cancer tissue compared to adjacent normal tissues [93,94]. The inhibition of KLF2 expression results from increased levels of long non-coding RNA IRAIN (lncRNA IRAIN), lncRNA SNHG15, and lncRNA DUXAP8 [94,95,96]. LnRNA IRAIN and lncRNA DUXAP8 are shown to recruit histone demethylase lysine-specific demethylase 1 (LSD1) and an enhancer of zeste homolog 2 (EZH2) and lnRNA SNHG15–EZH2, leading to induction of histone H3 lysine 27 trimethylation (H3K27me3) or demethylation of H3K4, resulting in repression of *KLF2*, cyclin-dependent kinase inhibitor 1A (*CDKN1A*), and cyclin-dependent kinase inhibitor 2B (*CDKN2B*) genes, which ultimately causes increased PDAC proliferation [94,95,96]. Knockdown of *KLF2* showed increased PDAC metastasis. Conversely, its overexpression in pancreatic cancer cell lines led to inhibition of proliferation and migration, reduced colony formation efficiency compared to control cells, and decreased expression of β-catenin/TCF4 targets such as Jun proto-oncogene (*JUN*), MYC proto-oncogene (*MYC*), cyclin D1 (*CCND1*), and snail family transcriptional repressor 1 (*SNAI1*) [93]. KLF2 has been shown to interact with β-catenin directly, and by sequestration, it inhibits the transcriptional activity of the β-catenin/TCF4 complex [93]. Another study showed that reducing levels of *CDK8*, known as an oncogene, and *CTNNB1*, led to an increase in KLF2 levels and its transcriptional target semaphorin 3F (*SEMA3F*) and simultaneously decreased the levels of proteins stimulating angiogenesis such as vascular endothelial growth factor A (VEGF), vascular endothelial growth factor receptor 2 (VEGFR2), MMP9, C-MYC, and cyclin D1 [97]. In vivo studies show that the overexpression of *CDK8* has negative results on KLF2 expression and leads to an increase in tumor size and the presentation of endothelial markers [97]. Aside from reducing proliferation, KLF2 has been shown to induce senescence in PDAC cells. KLF2 formed a complex with forkhead box O4 (FOXO4), increasing p21 on mRNA and protein levels. Overexpression of *KLF2* resulted in more nominal growth of organoids, suggesting that it may impair the activity of pancreatic cancer stem cells, and xenografts studies showed that its overexpression reduced the tumorigenicity of pancreatic cancer [98].

#### 4.1.2. KLF3

MiRs negatively regulate KLF3 in PDAC. Inhibition of miR-324-5p or overexpression of *KLF3* in pancreatic cancer cells led to reduced proliferation and increased apoptosis, as shown by decreased levels of proliferating cell nuclear antigen (*PCNA*) and increased levels of BCL2-associated X (*BAX*), respectively [99]. Analysis with TargetScan identified potential binding sites for miR-324-5p of 3’UTR of *KLF3*; as a result, ectopic expression of miR-324-5p led to a reduction in KLF3 RNA and protein levels. In another study, M2 macrophage-derived exosomal miR-21-5p was shown to negatively regulate *KLF3* in pancreatic cancer stem cells and, thus, stimulate pancreatic cancer growth and proliferation. The miR-21-5p binding site was identified in 3’UTR of *KLF3*, confirmed with a dual luciferase assay [100]. In addition, miR-21-5p from M2 macrophage-derived exosomes reduced the levels of KLF3 in the pancreatic stem cells derived from pancreatic cancer cell lines. Overexpression of *KLF3* in these cells reduced the levels of known stem cell pluripotent markers such as NANOG and OCT4, inhibited spheres and colonies formation, invasion, and migration, and induced their apoptosis [100].

#### 4.1.3. KLF6

Similarly to KLF2 and KLF3, KLF6 is another tumor suppressor whose levels are inhibited in PDAC tissues and whose overexpression reduces pancreatic cancer proliferation, migration, and invasion [133]. These KLF6 functions are partially driven by the upregulation of activating transcription factor 3 (*ATF3*) that leads to an increase in E-cadherin and downregulation of N-cadherin, MMP2, and vimentin. It has been shown that the ATF/CREB family of transcription factors is involved in many cellular functions, including cell proliferation and metastasis [119]. In addition, KLF6 levels were shown to be downregulated in gemcitabine (GEM)-resistant PDAC tissues and cells compared to GEM-sensitive PDAC [120]. The study showed that downregulation of *KLF6* is mediated through miR-342-3p. Leptin binds to the leptin receptor and activates the NOTCH pathway and expression of miR-342-3p, which, in turn, causes cleavage of *KLF6* mRNA. Reduction in its transcription results in an increased GEM-resistant phenotype of pancreatic cancer cells [120].

#### 4.1.4. KLF9

KLF9 is another downregulated KLF in pancreatic cancer [124]. Overexpression of *KLF9* leads to reduced cell proliferation and clone formation in soft agar and reduces the activity of TOPFLASH reporter, while its knockdown has the opposite effect. Forced expression of *KLF9* leads to the downregulation of Frizzled class receptor 5 (*FZD5*) and, thus, inhibition of the WNT/β-catenin pathway. KLF9 increase counteracted the positive impact of WNT3a and disheveled segment polarity protein 2 (DVL2) on WNT/β-catenin activity [124]. Another study confirmed the negative effect of KLF9 on pancreatic cell proliferation. It showed that KLF9 could induce apoptosis, block the cell cycle at the S phase, and inhibit the migration and invasion of tumor cells [125]. *KLF9* overexpression led to downregulating MMP9, MMP2, BCL2 apoptosis regulator (BCL2), N-cadherin, and cyclin B, and upregulating E-cadherin, BAX, TP53, cyclin-dependent kinase 4 (CDK4), and cyclin D1 [125].

#### 4.1.5. KLF10

KLF10 is expressed in the exocrine compartment of a normal pancreas (acinar and ductal cells) and is downregulated in PDAC [134]. During pancreatic carcinogenesis, DNA methyltransferase 1 (DNMT1) methylation of *KLF10* promoter contributes to its loss [135]. The decrease in *KLF10* leads to enhanced proliferation, migration, and invasion of pancreatic cancer. Another study showed that KLF10 levels in pancreatic cancer can be regulated by degradation [134]. It has been shown that peptidylprolyl cis/trans isomerase, NIMA-interacting 1 (PIN1), and Raf-1 proto-oncogene (RAF1) regulate the stability of KLF10 via the phosphorylation of threonine 93, leading to its degradation [134].

Furthermore, FLVCR1 antisense RNA 1 (FLVCR1-AS1) regulates *KLF10* expression through the sponging of miR-513c-5p and miR-514b-5p [126]. This may suggest that miR-513c-5p and miR-514b-5p are negative regulators of *KLF10* expression. On the other hand, in a positive feedback loop, KLF10 can stimulate the expression of FLVCR1-AS1. In conclusion, overexpression of FLVCR1-AS1 in pancreatic cancer cell lines increases cell cycle arrest and reduction in proliferation, migration, and wound healing [126].

KLF10 is a mediator of TGFβ signaling [136]. In the TGFβ-sensitive pancreatic cancer cell line, KLF10 levels were induced by TGFβ treatment and resulted in cell apoptosis [137]. Interestingly, even in TGFβ -resistant cells, *KLF10* overexpression led to apoptosis, and, notably, it increased sensitivity to gemcitabine treatment in PDAC cell lines [138]. Multiple studies in vitro and in vivo showed that ectopic expression of *KLF10* led to increased apoptosis and delayed growth of orthotopic tumors, while *Klf10* deletion led to aggressive tumor growth. Deletion of *Klf10* in *Pdx-1^Cre^LSL-Kras^G12D^Klf10^fl/fl^* mice resulted in significantly higher activation of TGFβ, MAPK, PI3K, NOTCH, and WNT/β-catenin pathways compared to *Pdx-1CreLSL-Kras^G12D^* mice [127]. Furthermore, its deletion in the setting of mutated Kras and deleted p53 (*Pdx-1^Cre^LSL-Kras^G12D^p53^fl/fl^Klf10^fl/fl^* mouse model) stimulated an invasive, metastatic phenotype to organs such as the stomach, lung, and liver [127]. KLF10 loss prompted activation of C-X-C motif chemokine ligand 12 (CXCL-12), and PDAC null for KLF10 showed increased staining of CXCL-12 and its partner, C-X-C motif chemokine receptor 4 (CXCR4), compared to the controls [127]. Microarray analysis demonstrated that CXCL-12/CXCR4 and AP-1 pathways are significant mediators of loss of KLF10 in pancreatic tumorigenesis. Using Plerixafor, a CXCR4 inhibitor, inhibited PDAC formation in *Pdx-1^Cre^LSL-Kras^G12D^Klf10^fl/fl^* and *Pdx-1^Cre^LSL-Kras^G12D^p53^fl/fl^Klf10^fl/fl^* mice, overriding the effects of *Klf10* deletion. Additionally, in pancreatic cancer cells, upon *KLF10* deletion, activation of the TGFβ and WNT pathway was observed by increased levels of active-β-catenin, c-MYC, survivin, cyclin D1, and Twist family bHLH transcription factor 2 (TWIST2), and stem cell markers such as CD133, CD44, nestin, and ATP-binding cassette subfamily G member 2 (ABCG2) [127].

In pancreatic cancer cells, glycolysis is a primary energy source to sustain proliferation, migration, invasion, and metastasis [139]. Deletion of *KLF10* in pancreatic cancer led to downregulation of glucose uptake, lactate production, and glycolytic activity and reduction in mitochondrial oxidative phosphorylation, basal respiration, and maximal respiration capacity [128]. Sirtuin 6 (SIRT6) is a direct mediator of KLF10 in EMT and glucose metabolism, and its overexpression reverted phenotypes connected with *KLF10* deletion [128].

In PDAC treatment, response to radiation therapy plays an important role. A comparison of biospecimens from patients with good responses to chemotherapy showed increased levels of KLF10 and lower levels of UV radiation resistance-associated gene (*UVRAG*) [140]. KLF10 has been shown to bind to the promoter of *UVRAG* and repress its transcription. In vitro and in vivo studies showed that treatment with metformin induced *KLF10*, reduced *UVRAG,* and sensitized cells to radiation injury. The same study demonstrated that KLF10 can regulate radiosensitivity of pancreatic cancer cells by modulating autophagy [140]. Overexpression of *KLF10* in MIA PaCa-2 cells combined with irradiation led to decreased levels of autophagy-related proteins, increased apoptosis, and elevated DNA damage, while its knockdown had the opposite effects. Taken together, these studies show that loss of KLF10 activates multiple pathways, accelerates the development of PDAC, and plays an important role in the response to current therapies.

#### 4.1.6. KLF11

KLF11 is another mediator of TGFβ signaling, regulating growth inhibition in untransformed epithelial cells [141]. KLF11 is expressed in the normal pancreas; however, in pancreatic cancer cells, its expression is heterogenous. It was shown that SMAD3 cooperates with KLF11 to repress *c-MYC* transcription by binding it to its translation inhibitory element (TIE) core. In pancreatic cancer cell lines bearing *Kras* mutation, the interaction between KLF11 and SMAD3 is inhibited, and treatment with ERK pathway inhibitors such as U0126 (highly selective inhibitor of mitogen-activated protein kinase kinase 1/2–MEK1/2) can revert the Kras effect. In addition, KLF11 was shown to repress TGFβ-induced transcription from the *SMAD7* promoter by recruiting SIN3 transcription regulator family member A (SIN3a) to GC-rich sites [129]. However, in pancreatic cancer, oncogenic KRAS activates the ERK pathway, which, in turn, phosphorylates KLF11, and abrogates KLF11/SIN3a interaction, leading to transcriptional activation of *SMAD7* and tumor progression.

#### 4.1.7. KLF13

KLF13 was confirmed to inhibit the EMT process of pancreatic cancer by promoting the transcription of LINC00261 and suppressing the expression of metastasis-associated proteins, thus, inhibiting the metastasis of pancreatic cancer [131]. It was shown that overexpression of LINC00261 inhibited migration and invasion of pancreatic cancer cells but did not induce apoptosis. The effect on EMT was demonstrated by upregulation of epithelial markers such as E-cadherin and downregulation of mesenchymal markers vimentin and MMP2 but not MMP9 or MMP14. Moreover, overexpression of *KLF13* could inhibit the phosphorylation of mTOR, ribosomal protein S6 kinase B1 (P70S6K1), and S6 proteins, and silencing of LINC00261 reversed this effect [131]. The PI3K/AKT/mTOR signaling pathway upregulates the mRNA and protein expression of MMP2, which could degrade the extracellular matrix to promote the metastasis of tumor cells. Thus, the sequence of events during pancreatic cancer tumorigenesis is upregulation of *KLF13*, resulting in increased levels of *LINC00261* and a decrease in PI3K/AKT/mTOR pathway activity, leading to a decline in metastasis [131].

Taken together, multiple KLFs function as tumor suppressors in PDAC development and exert their roles by modulating the expression of various pathways important for cancer development. Notably, as in the case of KLF6 and KLF10, they have been shown to play an important role in sensitizing PDAC cells to radio and chemotherapy.

### 4.2. KLFs as Oncogenes

#### 4.2.1. KLF7

*KLF7* is overexpressed in PDAC due to activation of a MAP kinase pathway or inactivation of the tumor suppressor p53, two alterations that occur in most pancreatic cancers [121]. TP53 regulates the expression of *KLF7* in PDACs by directly binding its promoter and blocking its transcription. *KLF7* knockdown results in decreased expression of IFN-stimulated genes (ISGs) such as interferon-induced protein with tetratricopeptide repeats 1 and 3 (*IFIT1*) and (*IFIT3*) and downregulation of discs large MAGUK scaffold protein 3 (*DLG3*). The first pathway is necessary for KLF7-mediated PDAC tumor growth and metastasis, and the latter causes Golgi complex fragmentation and results in reduced protein glycosylation, leading to reduced secretion of chemokines [121]. *KLF7* knockdown in vitro and in vivo reduces pancreatic cancer cell proliferation and tumor growth, respectively. Furthermore, it was shown that LINC 00152 regulates glycolysis, and LINC 00152/miR-185-5p complex negatively regulates *KLF7* and glycolysis in PDAC cells [122]. The study showed that changes to the levels of glycolysis-related enzymes such as hexokinase 2 (HK2), 6-Phosphofructo-2-Kinase/Fructose-2,6-Biphosphatase 3 (PFKBF3) and pyruvate dehydrogenase kinase 1 (PDK1) were observed upon downregulation of KLF7. It was demonstrated that miR-185-5p directly binds to 3′UTR of *KLF7* and reduces its expression levels. Thus, upregulation of KLF4 in PDAC positively impacts pancreatic cancer progression by promoting glycolysis and facilitating angiogenesis and EMT [122].

#### 4.2.2. KLF8

KLF8 levels are increased in PDAC cell lines and in the tissues originated from xenograft studies of PDAC [123]. Knockdown of *KLF8* resulted in the downregulation of cyclin-dependent kinase 1 (CDK1/CDC2), cyclin B1, and cyclin D1 proteins and upregulation of p21 and p27, leading to reduced cell proliferation and induction in G2/M phase arrest. Aside from positively regulating pancreatic cells’ expansion, KLF8 was shown to positively promote EMT via transcriptional activation of four and a half LIM domains 2 (FHL2), a mediator of snail family transcriptional repressor 2 (SNAI2) activation, and repression of E-cadherin [123].

#### 4.2.3. KLF12

KLF12 is increased in pancreatic cancer specimens compared to normal adjacent tissue and positively regulated by circ_0005273 and LncRNA-PACERR [142,143]. Knockdown of circ_0005273 reduced the proliferation and migration of pancreatic cell lines, mediated by KLF12. It has been shown mechanistically that lncRNA-PACERR activates the KLF12/p-AKT/c-MYC pathway by sponging miR-671-3p in tumor-associated macrophages (TAMs) [143]. LncRNA-PACERR bound to insulin-like growth factor 2 mRNA-binding protein 2 (IGF2BP2) acted in an m6A-dependent manner and enhanced the stability of KLF12 and c-MYC in the cytoplasm. In the meantime, KLF12 promoted the transcription of lncRNA-PACERR, forming the KLF12/lncRNA-PACERR complex that recruited histone acetyltransferase-EP300 to lncRNA-PACERR promoter. KLF12-positive TAMs increase pancreatic cancer cell proliferation, migration, and invasion [143]. On the other hand, KLF12 is a target of and negatively regulated by miR-137, a tumor suppressor in PDAC [130]. Overexpression of miR-137 decreased the proliferation and invasiveness of pancreatic cancer cells. Ectopic expression miR-137 reduced the population of CD133-positive cells, which correlated with downregulated levels of multiple stemness markers such as B lymphoma Mo-MLV insertion region 1 homolog (BMI1), leucine-rich repeat containing G protein-coupled receptor 5 (LGR5), NANOG, OCT4, and SOX2 on both mRNA and protein levels. Likewise, *KLF12* knockdown led to similar results [130]. In pancreatic cancer specimens, the levels of KLF12 positively correlated with genes activated by the WNT/β-catenin pathway [130]. KLF12 binds β-catenin and prevents its phosphorylation at serine 45 and inhibits degradation, while miR-137 represses KLF12-mediated β-catenin transcriptional activity. Furthermore, this study showed that KLF12 positively regulated *DVL2*, a repressor of the APC-mediated degradation pathway, and activated the expression of β-catenin-mediated genes such as *C-MYC*, *CCND1*, *TCF4*, *CD44*, *MMP7*, and *TWIST1* [130].

#### 4.2.4. KLF16

Previously, KLF16 has been shown to play a pro-tumorigenic role in various cancers such as breast, prostate, and gastric [144,145,146,147]. Analysis using gene expression profiling interactive analysis (GEPIA) shows that KLF16 is highly expressed in PDAC compared to normal adjacent tissue and its expression is positively correlated with SMAD6 expression [132]. An in vitro study confirms that KLF16 positively regulates PDAC cells proliferation, cell cycle, and migration. Transcriptomic analysis shows that downregulation in *KLF16* results in a decrease in the genes related to ribosomal pathway and an increase in autophagy, and that SMAD6 mediates KLF16 function in PDAC [132].

### 4.3. KLFs with a Biphasic Role in PDAC

#### 4.3.1. KLF4

KLF4 is universally known as a tumor suppressor [148,149]. One of the first studies shows that KLF4 positively regulates the expression of keratin 19 when overexpressed in pancreatic acinar cells [102]. The follow-up investigations show that KLF4 and KRT19 expression is positively correlated in pancreatic ductal cells and that KLF4’s and several other foregut markers’ expression is increased in early pancreatic neoplasia compared to the normal ducts [101,103]. In the context of PDAC, KLF4 levels are increased in cancer stem cells alongside three other Yamanaka factors: OCT4, NANOG, and SOX2 [150]. On the other hand, it has been shown that downregulation of *KLF4* gene expression is an early event in PDAC progression and is associated with loss-of-heterozygosity at 9q22.3–32 in PDAC and in PanIN [151]. Whole exome sequencing of non-invasive cyst-forming neoplasm called intraductal papillary mucinous neoplasms (IPMNs) identified hotspot mutations in zinc finger domains of *KLF4* gene in more than half of IPMNs [152]. The four more in-depth analyzed mutations were classified as drivers of IPMNs tumorigenesis according to CHASMplus software that allows for the identification of driver missense mutations [152,153]. KLF4 expression is significantly decreased in pancreatic cancer and correlates with the invasion and pancreatic disease stage [110]. It has been shown that overexpression of *DNMT1* leads to *KLF4* promoter hypermethylation, which contributes to decreased expression and is associated with poor differentiation of pancreatic cancer [154]. In the same study, the authors showed that DIM significantly induced miR-152 expression, blocking DNMT1, its binding to *KLF4* promoter, thus, activating *KLF4* expression, and inhibiting cell growth in vitro and tumorigenesis in animal models of pancreatic cancer [154]. KLF4 is a downstream target of miR-135b-5p [104]. miR-135b-5p is an oncogene that negatively regulates *KLF4* expression by direct binding to its 3′UTR region, resulting in reduced levels of *KLF4* and activation of G protein-coupled receptor class C group 5-member A (GPRC5A), and promoting the malignant progression of pancreatic cancer [104]. 

KLF4 plays an essential role in the initiation of pancreatic cancer development. KLF4 levels are upregulated during injury, ADM, and early pancreatic neoplasia, and its deletion attenuates the formation of pancreatic intraepithelial neoplasia induced by mutant Kras^G12D^ [105,155]. AP increases expression levels of activating transcription factor 4 (*ATF4*), which transcriptionally activates histone deacetylase 1 (*HDAC1*), inhibiting NEP expression and leading to increased levels of *KLF4*. Overexpression of *KLF4* resulted in increased expression of epithelial (ductal) markers such as KRT19 and diminished expression of acinar markers [105,155]. Secreted mucin 5AC (MUC5AC), a marker for early pancreatic neoplasia, regulates the development of pancreatic cancer and promotes the expression of cancer stem cell markers in an autochthonous murine model of pancreatic cancer (*Pdx-1cre*, *KrasG12D*) [156]. Deletion of *Muc5ac* in these mice resulted in decreased levels of aldehyde dehydrogenase 1 family member A1 (*Aldh1a1*), *Klf4*, epithelial cell adhesion molecule (*EpCAM*), and *CD133*. MUC5AC stimulates signaling through integrin avb5, pSrc (Y416), and pSTAT3 (Y705), which upregulates KLF4 and increases the tumorigenic propensity of cancer stem cells [156]. Alcohol is one of the factors boosting the chance of pancreatic cancer development [157]. Chronic alcohol treatment led to EMT in vitro and promoted cancer development in *Pdx-1^Cre^*, *Kras^G12D^* mice. Pancreas isolated from *Pdx-1^Cre^*, *Kras^G12D^* mice fed with an ethanol-containing diet showed higher expression of stem cell markers (CD133, CD44, CD24), pluripotency-maintaining factors (cMYC, KLF4, SOX2, and OCT4), EMT markers (N-cadherin, SNAI1, SNAI2, and zinc finger E-box-binding homeobox 1 (ZEB1)), and lower expression of epithelial markers such as E-cadherin than those isolated from mice fed with a control diet [157].

Wei and colleagues showed that KLF4 could exist in four isoforms in pancreatic cancer because of cis-splicing [158]. Interestingly, the KLF4 alpha isoform lacks nuclear localization signal, resides in the cytoplasm, and acts as an oncogene [158]. Ectopic expression of *KLF4* alpha form reduced the expression of *CDKN1A* and *CDKN1B*, promoting cell cycle progression and in vivo tumor formation. In addition, they show that KLF4 alpha competes with wild-type (WT) KLF4 for DNA binding and enforces its cytoplasmic localization [158]. To this end, overexpression of WT *KLF4* blocked cells in the G1 phase, reduced the number of cells in G2/M and S-phases, induced significant decrease in the proliferation associated with upregulation of *CDKN1A* and *CDKN1B,* and the downregulation of *CCND1* and S-phase kinase-associated protein 2 (*SKP2*) [159]. Furthermore, loss of KLF4 during PDAC progression increased Musashi-2 (MSI2) expression, promoted PDAC proliferation, migration, and invasion in vitro, and growth and metastasis in vivo [106]. In contrast, the knockdown of MSI2 expression caused the opposite.

Deletion of *ZEB1* in PDAC led to the development of epithelial phenotype and increased *KLF4* and E-cadherin (*CDH1*) expression [107]. Furthermore, tumors derived from PDAC cell lines with ZEB1 inhibition grew smaller and developed fewer metastases. Another study showed positive correlation between KLF4 and E-cadherin, while KLF4-negative PDAC tissues were, for the majority, positive for vimentin, a mesenchymal marker [110]. In vitro and in vivo studies show that downregulation of *KLF4* leads to increase in proliferation, migration, and invasion of the AsPC-1 PDAC cell line [110]. Notably, the decrease in *KLF4* expression is correlated with increased expression of caveolin-1 (*Cav-1*), and a study by Zhu and colleagues shows that KLF4 directly inhibits cav-1 expression [110]. The loss of KLF4 during the progression of PDAC and metastasis results in specificity protein 1 (SP1) binding to the forkhead box M1 (*FOXM1*) promoter, increasing its transcription [160]. FOXM1 induces pancreatic cancer cell growth, migration, and invasion by increasing the expression of *CCNB1*, *CCND1*, *CDK2*, *MMP2*, *MMP9*, and *VEGF,* and activates mesenchymal cell markers promoting EMT. MiR-10b enhanced the stimulatory effects of EGF and TGFβ on cell migration and EMT, and decreased the expression of KLF4 in PDAC cell lines [161]. These results show that downregulation of KLF4 during pancreatic cancer progression results in increased cell migration, invasion, and metastasis.

The miR-143/145 cluster regulates KLF4 and other pluripotency markers alongside KRAS [162]. It has been shown that doublecortin-like kinase 1 (DCLK1) plays an important role during pancreatic tumorigenesis by negatively regulating a set of miRs with a tumor-suppressing function such as miR-145, miR-200a, b, c, and let-7 [162]. Notably, PDAC with mutated KRAS has a loss of miR-145 expression, leading to an increase in the expression of pluripotency markers in cancer stem cells. Treatment of pancreatic cancer cell line with XMD8-92, a kinase inhibitor, led to similar effects as downregulation of DCLK1, upregulation of miR-143/145, miR-144, miR-200a, b, c, and let-7, and a decrease in the expression of proliferative, metastatic, and angiogenic markers (*c-MYC*, *KRAS*, *NOTCH1*, *ZEB1*, *ZEB2*, *SNAIL*, *SLUG*, *OCT4*, *SOX2*, *NANOG*, *KLF4*, *LIN28*, *VEGFR1*, and *VEGFR2*) [163]. Several studies reveal that cancer stem cell subpopulations marked by c-MET or CD24/CD44/EpCAM expression are characterized by high expression of reprogramming factors such as KLF4 [164,165]. Contrary to the studies above, Yan and colleagues show that KLF4 and CD44 expression is mutually exclusive, specifically within human metastatic pancreatic tumors and in autochthonous mouse models of PDAC [108]. They demonstrate that KLF4 directly binds to the *CD44* promoter and represses its transcription, reducing stemness and inhibiting tumorigenesis and metastasis. Another study supported KLF4’s self-renewal role in pancreatic cancer stem cells. CSCs obtained from primary patient-derived pancreatic cancer cells showed a positive correlation between higher telomerase activity and longer telomeres and stemness factors (NANOG, OCT3/4, SOX2, KLF4) [109]. Using genetic and chemical inhibition of telomerase, they show a positive feedback loop between telomerase and stemness factors maintaining self-renewal of CSC.

As we know, treatment of PDAC is ineffective, and patients rapidly develop resistance to standard therapies [166,167]. As such, investigations into the potential role of KLF4 in developing resistance to the treatment of PDAC have been pursued. One study showed that gemcitabine treatment reduced the expression of *KLF4*, miR-200b, and miR-183 but promoted *ZEB1* expression, suggesting an increase in the mesenchymal phenotype [107]. KLF4 overexpression promoted the expression of miR-200b and miR-183, suggesting that KLF4 positively regulated the expression of miR-200b and miR-183. This is by KLF4 promoting and maintaining the epithelial phenotype of PDAC. Metabolic reprogramming, such as “reverse Warburg,” announces a more aggressive phenotype and worse prognosis that is accompanied by increased stemness and reprograming factors such as KLF4, NANOG, and SOX2 [168].

#### 4.3.2. KLF5

KLF5 is the most studied KLF in pancreatic tumorigenesis. Its role has been shown in pancreatic injury, acinar-to-ductal metaplasia, and pancreatic cancer. The expression of KLF5 is high in embryonic pancreatic cells and decreases in the postnatal period [169]. Accordingly, KLF5 is expressed in pancreatic ductal cells but not in the acinar compartment during homeostasis. Analysis of tissues originated from mouse models of acute pancreatitis and specimen from pancreatitis patients shows that KLF5 is induced in pancreatic acinar cells and is crucial to ADM and the formation of early neoplasia [111]. Using human pancreatic cancer tissues, others show that KLF5 is significantly increased in cancer tissues compared to the normal adjacent pancreas [113]. Further analysis of low- and high-grade tumors and pancreatic cancer cell lines obtained from PDAC patients demonstrated that KLF5 was explicitly expressed in low-grade PDAC [112]. Multiple studies show that KLF5 has pro-tumorigenic effects in PDAC. The high throughput of pancreatic cancer cells using shRNA identified 185 genes crucial to pancreatic cancer growth [170]. Among others, KLF5 has been amplified in pancreatic cancer, and its knockdown decreases the proliferation of pancreatic cancer cells. Numerous studies reported association-specific genome loci, including KLF5 and pancreatic cancer risk [171,172].

Using *Ptf1a1^CreERTM^Klf5^fl/fl^* and *Ptf1a1^CreERTM^LSL-Kras^G12D^Klf5^fl/fl^* mice in combination with acute injury to the pancreas mediated by cerulein, we show that KLF5 is necessary for ADM and early pancreatic neoplasia formation [111]. Deleting *Klf5* in the context of acute pancreatitis by itself or in the context of Kras mutations blocks ADM and mPanIN building, resulting in decreased *Krt19* and *Ccnd1* and upregulated levels of *Ndrg2*, which were correlated with the reduced level of STAT3 phosphorylation [111]. Furthermore, this study demonstrates that KrasG12D mutation substantially affects KLF5 levels and, thus, ADM and mPanIN development (Figure 4).

KLF5 had been identified as a master regulator of epithelial phenotype in pancreatic cancer [112,114,115]. As such, KLF5 is pivotal during pancreatic cancer development and progression. However, its expression is downregulated during EMT and pancreatic cancer invasion. Thus, at first, KLF5 plays the role of pro-tumorigenic factor, and its role switches as its presence inhibits pancreatic tumor dissemination. Here, we describe the mechanisms regulating KLF5 expression in pancreatic cancer and their significance on its function at different stages of PDAC.

KLF5 has been known as a significant target of the MAPK/KRAS/ERK pathway in colorectal carcinogenesis [173]. However, at first, this pathway seemed to not be responsible for KLF5 induction in pancreatic cancer [113]. Instead, IL1β and hypoxia increased the levels of KLF5. Hypoxia-inducible factor 1 subunit alpha (HIF1α) and KLF5 formed a complex that increased the expression of solute carrier family 2 member 1 (*SLC2A1*), leading to increased transport of glucose, aerobic glycolysis, pyruvate, and lactate production [113] (Figure 4). In the same study, the authors show that the downregulation of KLF5 results in reduced survivin and platelet-derived growth factor subunit A (PDGFα), leading to reduced proliferation and increased apoptosis. Another way that KLF5 promotes glycolysis of pancreatic cancer is through transcriptional activation of phospholipase A and acyltransferase 3 (*PLA2G16*) [116] (Figure 4). KLF5 was positively correlated with PLA2G16 expression in PDAC tumors with TP53 mutation. Biochemical assays show that mutant p53 interacts with KLF5 and regulates *PLA2G16*. Downregulation of *PLA2G16* led to a substantial decrease in glucose uptake and lactate production in pancreatic cancer cells [116]. In cells originating from the pancreatic mouse model, KLF5 levels were downregulated due the activity of inhibitors of the PI3K and MAPK pathways [111]. Its levels were negatively regulated by miR-145 in vitro and in vivo, leading to the inhibition of proliferation, invasion, and migration [174]. As already shown, KLF5 regulates the epithelial phenotype of PDAC and has a significant effect on cell growth. KLF5-activated genes include *CCND1*, E2F transcription factor a (*E2F1*), pancreas-associated transcription factor 1a (*PTF1A*) and RAD51 recombinase (*RAD51*) [175] (Figure 4). Upregulation of these genes led to G1/S progression, uncontrolled proliferation, increased survival, genomic instability, and failure to repair genes. Moreover, KLF5 activates expression of various miRs including miR-130b-3p, miR-15a-5p, miR-17-5p, miR-183-5p, miR-18a-5p, miR-200a-3p, miR-221-3p, miR-222-3p, miR-454-3p, miR-125b-5p, miR-146a-5p, and miR-24-3p, which leads to a decrease in *SMAD4*, *P53*, *CDKN2A*, and *BRCA2* expression, respectively [175] (Figure 4).

A few studies investigate the relationship between the TGFβ pathway and KLF5. The role of TGFβ is context-dependent and can switch between pro-tumorigenic and tumor suppressive signaling in pancreatic cancer [176,177]. In TGFβ-sensitive pancreatic cancer cells, TGFβ signaling induces SOX4, which shifts its role from pro-tumorigenic to inducer of apoptosis [117]. Without TGFβ signaling, KLF5 interacts with SOX4 and induces pro-tumorigenic phenotype. However, activation by TGFβ results in the induction of *SMAD4* and EMT markers, resulting in the inhibition of *KLF5*. Studies in an animal model of pancreatic carcinogenesis, *Pdx-1^Cre^LSL-Kras^G12D^Cdkn2a^fl/fl^Smad4^fl/fl^* mice, combined with acute pancreatitis, showed reduced apoptosis compared to SMAD4-positive mice [117]. In addition, mice with deleted *Smad4* expressed epithelial marker—E-cadherin. Transcriptomic analysis in *Pdx-1^Cre^LSL-Kras^G12D^Cdkn2a^fl/fl^* and *Pdx-1^Cre^LSL-Kras^G12D^Cdkn2a^fl/fl^Smad4^fl/fl^* identified twenty transcription factors lost during mesenchymal transformation, including *Klf5*. Both studies demonstrate that induction in SMAD expression leads to activation of EMT genes and repression of *Klf5* and *Cdh1*, and, therefore, loss of epithelial phenotype, which is necessary for PDAC invasion. To support this notion, another study showed a negative correlation between ZEB1 and KLF5 in PDAC [118]. As mentioned previously, KLF5 is expressed in low-grade PDAC and regulates its epithelial characteristics. A comprehensive evaluation of the effects of *KLF5* knockdown and overexpression in pancreatic cancer cell lines led to the identification of genes directly regulated by this TF. KLF5 positively regulates epithelial identity genes, including keratins, focal adhesion genes, genes encoding gap junction proteins, mucins, genes encoding transcription factors specific to classical PDACs, and negative genes associated with EMT and stemness.

### 4.4. KLFs as Potential Biomarkers

Genome-wide association studies (GWAS) have been performed to functionally screen for and evaluate high-risk genotypes associated with pancreatic cancer, specifically PDAC, which accounts for more than 90% of cases [178]. Several *KLF* genes have been identified as prognostic biomarkers for their strong correlation to PDAC late tumor stages, reoccurrence, and poor survival outcomes. Their presence and deregulation can serve as potential clinical predictors of PDAC prognosis and therapeutic targets for novel treatment strategies.

Upregulation of specific KLF proteins in PDAC has been reported, associating their positive proliferative modulation and synergistic interactions with higher PDAC risk and poorer survival. High expression of these factors has been validated through extensive tumor characterization and chromosomal analyses from PDAC cell lines, transgenic mice studies, and human PDAC patient tissues.

GWAS has identified chr13q22.1 as a common susceptibility locus in PDAC, where the *KLF5* gene lies. An analysis of 8000 genotyped PanC4 individuals found that *KLF5* was strongly associated with pancreatic cancer [172]. In addition, further characterization of the chr13q22.1 locus found a single nucleotide polymorphism (SNP) mutation in a gene desert closest to both *KLF5* and *KLF12* that was significantly correlated with a higher risk of pancreatic cancer [179]. Expression quantitative trait locus analysis revealed an association between the SNP and *DIS3*, another regulatory gene, which was downregulated in carriers of the SNP mutation [179]. Since both *KLF5* and *KLF12* were found near this cascade of gene interactions and high-risk mutations, there is a possibility that these transcription factors play a role in allele-specific protein binding that results in variant SNP activity [179]. A later study validated this by identifying 12 genes, including *KLF5*, among 20 PDAC susceptibility loci. *KLF5* was found to be on the same locus as *DIS3*, which was previously shown to be a risk factor, suggesting *KLF5* may contribute to increased PDAC risk, as it is already known to be overexpressed in pancreatic tumorigenesis [180]. Data from 175 PDAC cases found nine genes highly expressed in PDAC tissue and were considered useful prognostic biomarkers [181]. Among this analysis, *KLF5* was found to be associated with lower disease-free survival probability and time [181]. High expression of KLF8 was also associated with poor PDAC prognosis and higher tumor stages [182]. Cytoplasmic positive expression in more than half of the 68 PDAC cases indicated a potential prognostic value. Kaplan–Meier analysis showed that patients who were negative for KLF8 had better prognoses [182].

Research on the cancer transcriptome has sought to narrow our understanding of *KLF5* association with PDAC found in genome-wide analyses. In one study that functionally screened for pancreatic cancer gene candidates, *KLF5* transcript levels were locally amplified in pancreatic cancer cell lines. The subsequent knockdown of those levels reduced cell growth in vitro [183]. This growth dependency suggests *KLF5* is an oncogene that drives cancer progression. High expression of KLF5 was also found in short-surviving PDAC patients and associated with shorter overall and tumor-free survival times [175]. At a transcriptional level, *KLF5* mRNA expression was nearly 31 times higher in shorter surviving patients (less than 1 year post-PDAC surgery) compared to longer surviving patients (5 years) [175]. When assessing low and high-grade PDAC tumors, *KLF5* mRNA and protein expression consistently increased at higher grades, indicating an association with greater cancer aggressiveness and poor prognosis [118]. When *KLF5* was knocked down in *Kras^G12D^* mice, PDAC progression significantly decelerated, suggesting a multifactorial regulation in cancer cell proliferation and survival [118]. Knockdown of *KLF5* in pancreatic cancer cell lines led to cell cycle arrest at the G1 phase, inhibiting cell DNA replication and division [175].

From an immune response perspective, KLF5 negatively correlates with B-cell naïve, CD8+ T cells, and macrophage MO cells, suggesting that overexpression in the tumor microenvironment inhibits the immune system’s ability to fight cancer [181].

Circulating tumor cells (CTCs) in the peripheral blood system have been shown to express KLF protein activity and are associated with poor prognosis in pancreatic cancer [184]. CTCs are the main driver of metastasis from the primary tumor site, as they travel and lodge themselves onto uninvolved tissues [185]. Therefore, CTCs with EMT phenotype can be a valuable biomarker of poor PDAC prognosis and risk of cancer reoccurrence [184]. KLF8 expression is a known driver of EMT and has been implicated in metastatic progression and outcome. One study isolating and analyzing CTCs from pancreatic cancer patients found that nearly 66% of CTCs were positive for KLF8 and 65% were positive for vimentin. This fibroblast intermediate is also an important biomarker of EMT in tumorigenesis [184]. CTC-positive patients had much higher rates of cancer reoccurrence after a year of follow-up and a shorter recurrence-free survival rate [184]. High levels of KLF8+/vimentin+ CTCs indicate poor prognostic outcomes and may serve as a biomarker for distant metastasis [184].

Although alternative splicing is an essential element of gene expression and capacity for various protein production, a disturbance in the balance of factors that regulate splice site selection has been shown to cause disease [186]. *KLF6* is a tumor-suppressor gene that reduces cancer cell growth in culture and tumors in xenograft mice [187]. One study correlated overexpression of alternative splice isoforms of *KLF6*, without *KLF6* mutation, with PDAC tumor grade and survival. These *KLF6* alternative splice variants were significantly increased in tumors and PDAC cell lines [187]. Although the mechanisms of generating *KLF6* splice variants in cancer are not fully understood, an association exists between these splice variants’ mRNA expression and PDAC tumor grade.

Consequently, KLF6 is found at lower molecular weights in carcinoma tissue, suggesting that splice isoforms inhibit the production of the wild-type full-length protein and its associated transcriptional functions [187]. Through further transcriptome analysis of clinical PDAC data, KLF6 splice isoform mRNA expression was positively correlated with tumor grade and survival outcome [187]. In poorly differentiated tumors, the KLF6 wild-type/splice isoform (wt/sp) ratio was lower than in well-differentiated tumors, indicating an increase in KLF6 isoforms in later PDAC stages. A wt/sp lower than 2 was associated with shorter survival (10 months), compared to the survival time of a patient with a wt/sp above 2 (21 months) [187].

Conversely, KLF7 has been shown to participate in the critical splicing control of several genes associated with survival in PDAC [188]. Transcription factor enrichment analysis identified KLF7 as a significant factor highly related to alternative splicing promoting survival [188]. KLF7 direct binding to splicing gene promoters, regulation of splice site kinetics, and recruitment of splicing components may have positively impacted PDAC survival. However, the molecular mechanism remains unknown [188]. KLF6 isoform and KLF7 expression can be biomarkers to better understand splicing pattern influence and subsequent survival outcomes in clinical PDAC.

KLF4 expression in malignant and benign pancreatic tissues was assessed via immunohistochemical staining, which found significant downregulation of KLF4 in PDAC tissues compared to benign lesions and normal pancreatic tissues [189]. Kaplan–Meier survival analysis validated this by associating negative KLF4 expression, as an independent factor, with shorter PDAC survival and poor prognosis [189]. 

KLF9 interacts with co-repressors that inhibit cell proliferation, differentiation, and stemness [190]. It was downregulated in PDAC, according to a study that analyzed its prognostic value in 149 PDAC patients that underwent resection. Low nuclear expression of KLF9 via IHC staining was observed in 62% of PDAC tumor tissues, compared to high expression in only 38% of tumors [191]. That same proportion of low KLF9-expressed tumor tissues was further characterized as poorly differentiated and highly invasive [191]. When analyzing survival outcomes, patients with negative KLF9 expression in their PDAC tumor cells saw a marked decrease in mean survival time (14.17 months) relative to patients who were positive for KLF9 expression (23.95 months) [191].

Similar to KLF9, KLF10 interacts with various co-repressors to inhibit transcription. KLF10 has been shown to mediate growth inhibition in cancer through anti-proliferative and apoptotic effects, while also blocking signaling pathways that ultimately reverse tumorigenic phenotype [135]. The consensus is that high KLF10 expression is not only significantly correlated with better overall survival and lower tumor stages but can also be used to predict treatment efficacy for formulating therapeutic interventions in PDAC. One study investigating the connection between KLF10 expression and clinical features of PDAC found that mRNA expression was generally downregulated in primary tumor sites compared to surrounding normal pancreatic tissue [135]. Through multivariable analysis, KLF10 loss was found to be associated with advanced PDAC stages and distant metastasis, revealing that the loss of regulatory repressive activity allows for uncontrollable growth that enhances tumor cell invasion and migration [135]. A separate gene expression profile analysis stratified PDAC patients based on genetic risk profiles to utilize targeted molecular therapies. KLF10 was one of thirteen genes involved in the low-risk prognostic signature, relatively defined as improved prognosis and increased sensitivity to cancer treatment [192]. The high-risk cohort, devoid of low-risk genes, had lower CD8 T/B cells and a higher neutrophil count associated with PDAC proliferation and metastasis [192]. When high KLF10 expression was combined with chemoradiotherapy (CRT) treatment, patients had significantly better recurrence-free and overall survival times, indicating KLF10 positively influenced the benefits of adjuvant CRT [193]. Not only can KLF10 expression be used as a clinical prognostic and risk indicator for PDAC, but it can also serve as a potential therapeutic target to molecularly complement current cancer treatments.

## 5. Conclusions

KLFs have been significantly implicated in pancreatic cancer for their extensive roles in cell proliferation, differentiation, and apoptosis. As major transcriptional regulators of the cell cycle, they influence the uncontrollable growth and instability that drives cancer cell survival. The individual heterogeneity and range of functions among the KLF proteins can make their roles in cancer paradoxical, found as both suppressors and drivers of oncogenesis. Their context-dependent functions rely on distinctive levels of intratumoral cell-type differentiation that vary significantly between molecular subtypes of pancreatic cancer. Many regulatory proteins have been shown to modulate KLF expression and activity in the tumor microenvironment. As multiple KLFs participate in pancreatic tumorigenesis and play opposite roles, it would be informative to perform a comprehensive analysis of their function and the relation between them in low-grade and high-grade PDAC to assess the extent of their synergism and contribution to pancreatic cancer.

Advancements in understanding how proteins bind and induce the activity of critical proliferative factors such as KLF4 are crucial for establishing biomarkers for disease prevention and diagnosis. Computational profiling and statistical analysis of the pancreatic cancer genome combined with in vitro and in vivo experiments provided valuable insight into the genetic and protein mechanisms that drive it.

Given the significant advancements in our understanding of generating pure, differentiated populations of tissue-specific cells, research has paved a path for utilizing iPSCs in clinical applications ranging from disease modeling to autologous cell therapies. These foundational protocols enable controlled and replicable iPSC experiments to study the natural physiological interactions of organs and their associated diseases in vitro. Novel personalized molecular modeling and treatment strategies have the potential to provide a solution to the chronic deficiencies and limitations of donor tissue transplants for T1D and late diagnoses in PDAC. Future directions on establishing higher differentiation efficiencies and mitigating tumorigenic risks in vivo will be necessary.

## Figures and Tables

**Figure 1 ijms-24-08589-f001:**
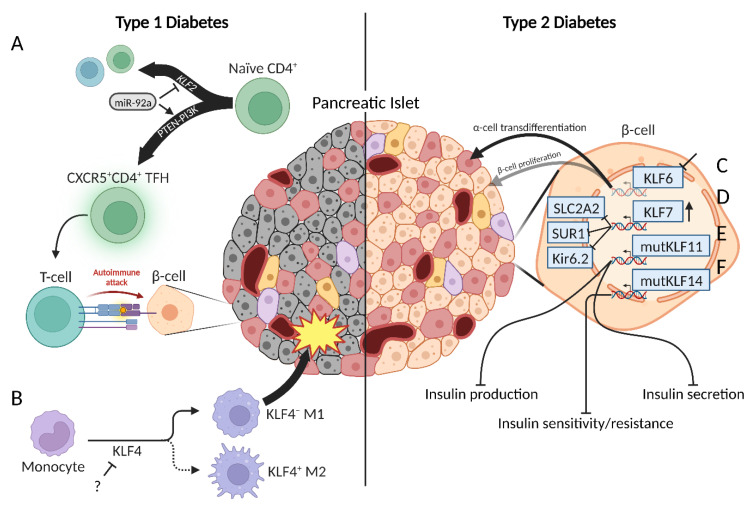
The regulatory role of KLFs in β-cell function in the setting of T1D (A,B) and T2D (C–F). (A) Inhibition of KLF2 by miR-92a mediates CXCR5^+^CD4^+^ TFH cells to enrich autoimmune T-cells. (B). Dysregulated KLF4 expression drives M1 macrophage subtype differentiation, in turn, promoting infiltration and inflammation. (C) KLF6 inhibition promotes β-cell de/transdifferentiation to glucagon-secreting α-cells. (D) Enhanced KLF7 expression inhibits transcription of proteins involved in cellular metabolism and insulin secretion. Mutations in KLF11 (E) and KLF14 (F) dysregulate insulin production and secretion and sensitivity/resistance, respectively. Created with BioRender.

**Figure 2 ijms-24-08589-f002:**
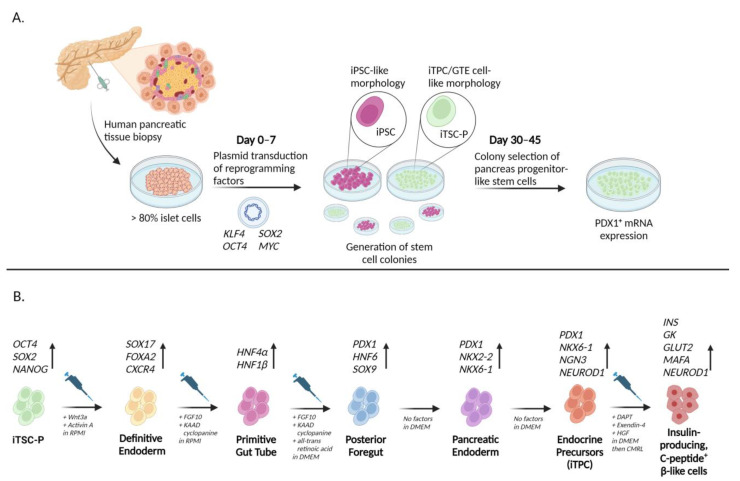
Generation of human pancreas-derived stem cells utilizing KLF4 to reprogram cell fate towards pluripotency with the presence of pancreatic cell lineage markers. (**A**). Colonies with pancreatic stem cell-like morphologies were selected for further experimentation including teratoma potential in mice and *PDX1* mRNA expression, a marker of pancreatic stem/progenitor cells. Colonies with no teratoma formation and high relative expression of *PDX1* were used for subsequent differentiation protocol to generate induced pancreatic progenitor cells. (**B**). Stepwise-induced differentiation to functional pancreatic β cells from iTSC-P. Emulation of in vivo embryonic pancreatic development through supplementation of endodermal factors, endocrine precursors, and media shown below each transition arrow. Expression of characteristic genetic markers at each stage of development was measured to ensure pancreatic lineage expression and displayed above respective stage. iPSC: Induced pluripotent stem cell; iTSC-P: Induced tissue-specific stem cell derived from pancreas; GTE: Gut tube endoderm; iTPC: Induced tissue-specific progenitor cells (pancreas-specific). Created with BioRender.

**Figure 3 ijms-24-08589-f003:**
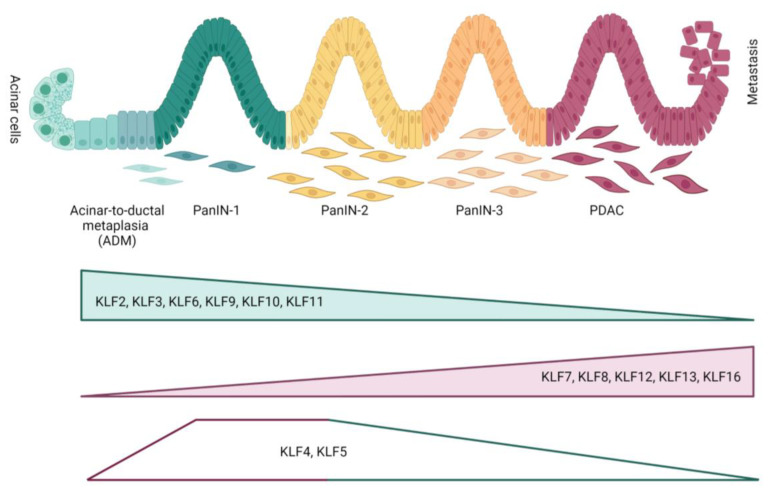
The pattern of KLFs expression during pancreatic carcinogenesis. For details, please see text and Table 1. Created with BioRender.

**Figure 4 ijms-24-08589-f004:**
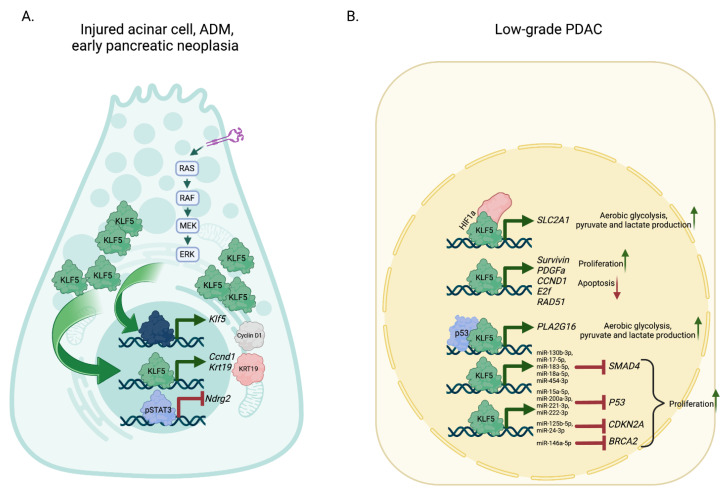
KLF5 regulates the development of ADM, early pancreatic neoplasia, and low-grade PDAC. (**A**). Upon injury to the pancreas and/or KRAS activation, KLF5 levels are upregulated, which, in turn, increases *Ccnd1*, *Krt19*, and phosphorylation of STAT3. As a result, there is an increase in proliferation of acinar cells and induction of ADM and PanIN formation. (**B**). In low-grade PDAC, KLF5 positively regulates proliferation, aerobic glycolysis, pyruvate, and lactate production, and reduces apoptosis. Created with BioRender.

**Table 1 ijms-24-08589-t001:** Cell types reprogrammed for iPSC generation and outcomes.

Initial Cell Type	Species	Induction Factors	Result	References
Dermal fibroblasts	Human	Yamanaka retroviral transduction, basic fibroblast growth factor supplementation	α- and β-like cells produced low levels of glucagon and c-peptide	[49]
Dermal fibroblasts from T1D patients	Human	KLF4, OCT4, SOX2 retroviral transduction, genetic/growth factor supplementation	Teratoma potential, embryoid bodies of cell types from all three germ layers produced, patient-specific disease modeling in vitro	[58]
Keratinocytes	Human	KLF4, OXT4, SOX2 retroviral transduction, growth factor supplementation	Higher reprogramming speed and efficiency compared to fibroblasts	[59]
In vivo	Mouse	rtTA, lentiviral doxycycline-inducible cassette encoding Yamanaka factors, DOX treatment	Well-differentiated teratomas emerged from multiple organs, highlighting diverse capabilities of in vivo iPSCs	[53]
Pancreatic cells	Mouse	Yamanaka retroviral transduction, genetic/growth factor supplementation	Expressed biomarkers of β-cell lineage, released insulin in response to glucose stimulation. Did not develop in-vivo teratomas	[56]
Pancreatic-specific progenitor cells	Human	Yamanaka and p53 shRNA transient expression, genetic/growth factor supplementation	Highest differentiation efficiency of insulin and c-peptide positive cells, highest levels of insulin released, lacked tumorigenic markers	[60]

**Table 2 ijms-24-08589-t002:** KLFs in pancreatic cancer.

Name	Status in Pancreatic Cancer	Downstream Targets	References
KLF2	Downregulated	Cyclin D1, JUN, MMP9, MYC, p21, SEMA3F, SNAI1, VEGF, VEGFR2	[93,94,95,96,97,98]
KLF3	Downregulated	BAX, NANOG, OCT4, PCNA,	[99,100]
KLF4	Stage-dependent	CD44, cyclin D1, E-cadherin, GPRC5A, KRT19, miR-183, miR-200b, MSI2, NANOG, OCT3/4, p21, p27, SKP2, SOX2, caveolin-1	[101,102,103,104,105,106,107,108,109,110]
KLF5	Stage-dependent	BRCA2, cyclin D1, E-cadherin, E2F1, KRT19, miR-130b-3p, miR-15a-5p, miR-17-5p, miR-183-5p, miR-18a-5p, miR-200a-3p, miR-221-3p, miR-222-3p, miR-454-3p, miR-125b-5p, miR-146a-5p, miR-24-3p, NDRG2, PDGFα, PLA2G16, PTF1α, p16, p53 RAD51, SLC2A1, SMAD4, survivin	[111,112,113,114,115,116,117,118]
KLF6	Downregulated	ATF3, MMP2, N-cadherin, vimentin	[119,120]
KLF7	Upregulated	DSG3, HK2, IFIT1, IFIT3, PDK1, PFKBF3	[121,122]
KLF8	Upregulated	CDK1/CDC2, Cyclin B1, cyclin D1, FHL2, p21, p27, SNAI2	[123]
KLF9	Downregulated	BAX, BCL2, CDK4, cyclin B, cyclin D1, E-cadherin, FZD5, MMP9, MMP2, N-cadherin, TP53	[124,125]
KLF10	Downregulated	ABCG2, CD133, CD44, CXCL-12, cyclin D1, FLVCR1-AS1, MYC, nestin, survivin, TWIST2	[126,127,128]
KLF11	Downregulated	c-MYC, SMAD7	[129]
KLF12	Upregulated	BMI1, CCND1, c-MYC, CD44, DVL2, LGR5, MMP7, NANOG, OCT4, SOX2, TCF4, TWIST1	[130]
KLF13	Upregulated	LINC00261, E-cadherin, MMP2, vimentin	[131]
KLF16	Upregulated	SMAD6	[132]

## Data Availability

Not applicable.

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
