# Peer review of "The Role of Krüppel-like Factors in Pancreatic Physiology and Pathophysiology"

_ijms, 2023, doi:10.3390/ijms24108589_

Round 1

Reviewer 1 Report

The manuscript summarizing the role of FLFs in pancreatic regeneration, inflammation, and carcinogenesis is well written and various aspects has been discussed in detail. There are few concerns to be addressed.

1. Section 2: Please include a schematic to show various KLFs regulating DMI and DMII involving beta-islets and their regulatory role (activating or inhibitory)

2.Section 3: Please include a Table summarizing the role of KLFs. Most of the text in this section is from references 48, 49, 52, 53, 55, 57, 58, 61, and 62. It will be better to summarize the text in brief and including a schematic. Further, Figure 1 depicts the methodology of developing iPSCs but KLFs have not been mentioned in the figure. Please include KLFs as discussed in the text. Else, the figure is a summary of what has been discussed in text. Section 3 is very long, please limit it using Tables and schematics.

Section 4: Well described but lack mentioning various recent articles describing the role of KLFs in PDAC based on sequencing studies.  KLFs play a critical role on various aspects of pancreatic physiology and pathology, and it is not possible to summarize all these aspects in one review. It will be better to focus either on physiological aspects of pathological aspects. Recent articles like these and others are missing.

https://www.ncbi.nlm.nih.gov/pmc/articles/PMC9694270/; https://pubmed.ncbi.nlm.nih.gov/33028669/; https://www.pnas.org/doi/10.1073/pnas.2005156117; https://www.karger.com/Article/FullText/488426; https://www.science.org/doi/10.1126/sciadv.abo3932;

Please expand all abbreviations on their first appearance.

Author Response

May 3rd, 2023

Editor and Reviewers

Dear Editor and Reviewers,

We want to thank you for your comments and suggestions. We hope that we have addressed all issues raised. Please see our responses to your comments below.

Reviewer 1.

Comment 1. Section 2: Please include a schematic to show various KLFs regulating DMI and DMII involving beta-islets and their regulatory role (activating or inhibitory)

Response 1. We provided a new figure (Figure 1) demonstrating KLFs' involvement in DMI and DMII.

Comment 2. Section 3: Please include a Table summarizing the role of KLFs. Most of the text in this section is from references 48, 49, 52, 53, 55, 57, 58, 61, and 62. It will be better to summarize the text in brief and including a schematic. Further, Figure 1 depicts the methodology of developing iPSCs but KLFs have not been mentioned in the figure. Please include KLFs as discussed in the text. Else, the figure is a summary of what has been discussed in text. Section 3 is very long, please limit it using Tables and schematics.

Response 2. According to the Reviewer’s suggestion, we shortened the main section of the text and included a table (Table 1) that summarizes approaches that utilize iPSCs. In addition, figure 2 (previous Figure 1) includes KLF4, as this factor was used for iPSCs.

Comment 3. Section 4: Well described but lack mentioning various recent articles describing the role of KLFs in PDAC based on sequencing studies.  KLFs play a critical role on various aspects of pancreatic physiology and pathology, and it is not possible to summarize all these aspects in one review. It will be better to focus either on physiological aspects of pathological aspects. Recent articles like these and others are missing.

https://www.ncbi.nlm.nih.gov/pmc/articles/PMC9694270/; https://pubmed.ncbi.nlm.nih.gov/33028669/; https://www.pnas.org/doi/10.1073/pnas.2005156117; https://www.karger.com/Article/FullText/488426; https://www.science.org/doi/10.1126/sciadv.abo3932;

Response 3. KLFs play an essential role in the physiology and pathophysiology of the pancreas. We understand that it is challenging to address the part of each KLF in minute detail fully. However, the submitted manuscript comprehensively reviews the current knowledge. We included publications suggested by the Reviewer, except the publication by Gupta et al., as it was included in the original submission.

Comment 4. Please expand all abbreviations on their first appearance.

Response 4. We provided full names to all abbreviations.

Reviewer 2.

We thank you, Reviewer 2, for your stellar and supportive comments regarding our manuscript.

We hope the incorporated changes will satisfy the Editor and Reviewers and render the revised manuscript suitable for publication. Thank you for being so considerate.

We confirm that neither the manuscript nor any parts of its content are currently under consideration or published in another journal.

All authors have approved the manuscript and agree with its submission to the Cancers.

Please feel free to contact me if I can be of further assistance,

Sincerely Yours,

Agnieszka B. Bialkowska, PhD

Associate Professor

Renaissance School of Medicine at Stony Brook University

Department of Medicine

GI Translational Research Lab

HSC-T17 Room 090

Stony Brook, NY 11794-8176

Phone: (631) 638 2161

Reviewer 2 Report

This is a comprehensive review article that describe KLFs’ function in maintaining homeostasis of the pancreas during the development and progression of pancreatic cancer and address their role in the regenerative process. The article is well written and can bring in-depth knowledge of KLFs to readers.

Author Response

(The authors gave the same response as above.)

Round 2

Reviewer 1 Report

None